# Poisson Process Jumping between an Unknown Number of Rates: Application to Neural Spike Data

**Florian Stimberg**
Computer Science, TU Berlin
Florian.Stimberg@tu-berlin.de

**Andreas Ruttor**
Computer Science, TU Berlin
Andreas.Ruttor@tu-berlin.de

**Manfred Opper**
Computer Science, TU Berlin
Manfred.Opper@tu-berlin.de

## Abstract

We introduce a model where the rate of an inhomogeneous Poisson process is modified by a Chinese restaurant process. Applying a MCMC sampler to this model allows us to do posterior Bayesian inference about the number of states in Poisson-like data. Our sampler is shown to get accurate results for synthetic data and we apply it to V1 neuron spike data to find discrete firing rate states depending on the orientation of a stimulus.

## 1 Introduction

Event time data is often modeled as an inhomogeneous Poisson process, whose rate $\lambda(t)$ as a function of time $t$ has to be learned from the data. Poisson processes have been used to model a wide variety of data, ranging from network traffic [25] to photon emission data [12]. Although neuronal spikes are in general not perfectly modeled by a Poisson process [17], there has been extensive work based on the simplified Poisson assumption [e.g. 19, 20]. Prior assumptions about the rate process strongly influence the result of inference. Some models assume that the rate $\lambda(t)$ changes continuously [1, 7, 22], but for certain applications it is more useful to model it as a piecewise constant function of time, which switches between a finite number of distinct states. Such an assumption could be of interest, when one tries to relate the change of the rate to sudden changes of certain external experimental conditions, e.g. changes of neural spike activity when external stimuli are switched.

An example for a discrete state rate process is the Markov modulated Poisson process (MMPP) [10, 18], where changes between the states of the rate follow a continuous time Markov jump process (MJP). For the MMPP one has to specify the number of states beforehand and it is often not clear how this number should be chosen. Comparing models with different numbers of states by computing Bayes factors can be cumbersome and time consuming. On the other hand, nonparametric Bayesian methods for models with an unknown number of model parameters based on Dirichlet or Chinese restaurant processes have been highly popular in recent years [e.g. 24, 26].

However—to our knowledge—such an idea has not yet been applied to the conceptually simpler Poisson process scenario. In this paper, we present a computationally efficient MCMC approach to this model, which utilizes its feature that given the jump process the observed Poisson events are independent. This property makes computing the data likelihood very fast in each iteration of our sampler and leads to a highly efficient estimation of the rate. This allows us to apply our sampler to large data sets.

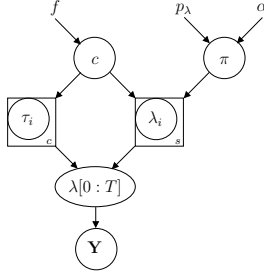

Figure 1: Generative model.

## 2 Model

We assume that the data comes from an inhomogeneous Poisson process, which has rate $\lambda(t)$ at time $t$. In our model $\lambda(t)$ is a latent, piecewise constant process. The likelihood of the data given a path $\lambda_{(0:T)}$ with $s$ distinct states then becomes [8]

$$P(\mathbf{Y}|\lambda_{(0:T)}) \propto \prod_{i=1}^{s} \lambda_i^{n_i} e^{-\tau_i \lambda_i}, \tag{1}$$

where $\tau_i$ is the overall time spent in state $i$ defined as $\lambda(t) = \lambda_i$ and $n_i$ is the number of Poisson events in the data $\mathbf{Y}$, while the system is in this state. A trajectory of $\lambda_{(0:T)}$ is generated by drawing $c$ jump times from a Poisson process with rate $f$. This means $\lambda_{(0:T)}$ is separated in $c + 1$ segments during which it remains in one state $\lambda_i$. To deal with an unknown number of discrete states and their unknown probability $\pi$ of being visited, we assume that the distribution $\pi$ is drawn from a Dirichlet process with concentration parameter $\alpha$ and base distribution $p_\lambda$. By integrating out $\pi$ we get a *Chinese restaurant process* (CRP) with the same parameters as the Dirichlet process. For a derivation of this result see [27].

Let us assume we already have $i$ segments and draw the next jump time from an exponential distribution with rate $f$. The next segment gets a new $\lambda$-value sampled from $p_\lambda$ with probability $\alpha/(\alpha + i)$, otherwise one of the previous segments is chosen with equal probability and its $\lambda$-value is also used for the new segment. This leads to the following prior probability of a path $\lambda_{(0:T)}$:

$$P(\lambda_{(0:T)}|f, \alpha, p_\lambda) \propto f^c e^{-fT} \alpha^s \frac{\prod_{j=1}^{s} (p_\lambda(\lambda_j)(\#_j - 1)!)}{\prod_{i=0}^{c} (\alpha + i)}, \tag{2}$$

where $s$ is the number of distinct values of $\lambda$. To summarize, we have $f$ as the rate of jumps, $p_\lambda$ as a prior distribution over the values of $\lambda$, $\#_j$ as the number of segments assigned to state $j$, and $\alpha$ as a hyperparameter which determines how likely a jump will lead to a completely new value for $\lambda$. If there are $c$ jumps in the path $\lambda_{(0:T)}$, then a priori the expected number of distinct $\lambda$-values is [28]

$$\mathcal{E}[s|c] = \sum_{i=1}^{c+1} \frac{\alpha}{\alpha + i - 1}. \tag{3}$$

We choose a gamma distribution for $p_\lambda$ with shape $a$ and scale $b$,

$$p_\lambda(\lambda) = \text{Gamma}(\lambda; a, b) \propto \lambda^{a-1} e^{-\lambda/b}, \tag{4}$$

which is conjugate to the likelihood (2). The generative model is visualized in figure 1.

## 3 MCMC Sampler

We use a Metropolis-within-Gibbs sampler with two main steps: First, we change the path of the Chinese restaurant process conditioned on the current parameters with a Metropolis Hastings random walk. In the seconds step, the time of the jumps and the states are held fixed, and we directly sample the $\lambda$-values and $f$ from their conditional posteriors.

## 3.1 Random Walk on the many-state Markov jump process

To generate a proposal path $\lambda^*_{(0:T)}$ (for the remainder of this paper $^*$ will always denote a variable concerning the proposal path) we manipulate the current path $\lambda_{(0:T)}$ by one of the following actions: shifting one of the jumps in time, adding a jump, removing one of the existing jumps, switching the state of a segment, joining two states, or dividing one state into two. This is similar to the birth-death approach, which has been used before for other types of MJPs [e.g. 5].

We **shift** a jump by drawing the new time from a Gaussian distribution centered at the current time with standard deviation $\sigma_t$ and truncated at the neighboring jumps. $\sigma_t$ is a parameter of the sampler, which we chose by hand and which should be in the same scale as the typical time between Poisson events. If in doubt, a high value should be chosen, so that the truncated distribution becomes more uniform.

When **adding** a jump the time of the new jump is drawn from a uniform distribution over the whole time interval. With probability $q_n$ a new value of $\lambda$ is added, otherwise we reuse an old one. The parameter $q_n$ was chosen by hand to be $0.1$, which worked well for all data sets we tested the sampler on.

To **remove** a jump we choose one of the jumps with equal probability.

**Switching** the state of a segment is done by choosing one of the segments at random and either assigning it to an existing value or introducing a value which was not used before, again with probability $q_n$.

When adding a new value of $\lambda$, both when adding a jump or when switching the state of a segment, we draw it from the conditional density

$$
\begin{aligned}
P(\lambda^*_{s+1}|\mathbf{Y}, \lambda_{(0:T)}) \quad &\propto \quad \mathrm{Gamma}(\lambda^*_{s+1}; a, b)\, \mathrm{Gamma}(\lambda^*_{s+1}; n_{s+1} + 1, 1/\tau_{s+1}) \\
&\propto \quad \mathrm{Gamma}\left(\lambda^*_{s+1}; a + n_{s+1}, b/(\tau_{s+1}b + 1)\right).
\end{aligned}
\tag{5}
$$

If we instead reuse an already existing $\lambda$, we choose which state to use by drawing it from a discrete distribution with probabilities proportional to (5), but with $n$ and $\tau$ being the number of Poisson events and the time in this segment, respectively.

Changing the number of states through adding and removing jumps or switching the states of segments is sufficient to guarantee that the sampler converges to the posterior density. However, the sampler is very unlikely to reduce the number of states through these actions, if all states are used in multiple segments, so that convergence might take a very long time in this case. Therefore, we introduce the option to **join** all segments assigned to a neighboring (when ordered by their $\lambda$-values) pair of states into one state. Here the geometrical mean $\lambda^*_j = \sqrt{\lambda_{i_1}\lambda_{i_2}}$ of both $\lambda$-values is used for the joined state.

Because we added the join action, we need an inverted action, which **divides** a state into two new ones, in order to guarantee reversibility and therefore fulfill detailed balance. The state to divide is randomly chosen among the states which have at least two segments assigned to them. Then a small factor $\epsilon > 1$ is drawn from a shifted exponential distribution and the $\lambda$-value of the chosen state is multiplied with and divided by $\epsilon$, respectively, to get the $\lambda$-values $\lambda^*_{j_1} = \lambda_i \epsilon$ and $\lambda^*_{j_2} = \lambda_i/\epsilon$ of the two new states. The distribution over $\epsilon$ is bounded, so that the new $\lambda$-values are assured to be between the neighboring ones. After this, the segments of the old state are randomly assigned to the two new states with probability proportional to the data likelihood (1). If by the last segment only one of the two states was chosen for all segments, the last segment is set to the other state. This method assures that every possible assignment (where both states are used) of the two states to the segments of the old state can occur. Additionally, there is exactly one way for each assignment to be drawn allowing a simple calculation of the Metropolis-Hastings acceptance probability for both the join and the divide action. Figure 2 shows how these actions work on the path.

A proposed path $\lambda^*_{(0:T)}$ is accepted with probability

$$
p_{\mathrm{MH}} = \min\left(1, \frac{P(\mathbf{Y}|\lambda^*_{(0:T)})}{P(\mathbf{Y}|\lambda_{(0:T)})} \frac{Q(\lambda_{(0:T)}|\lambda^*_{(0:T)})}{Q(\lambda^*_{(0:T)}|\lambda_{(0:T)})} \frac{P(\lambda^*_{(0:T)}|f, \alpha, p_\lambda)}{P(\lambda_{(0:T)}|f, \alpha, p_\lambda)}\right).
\tag{6}
$$

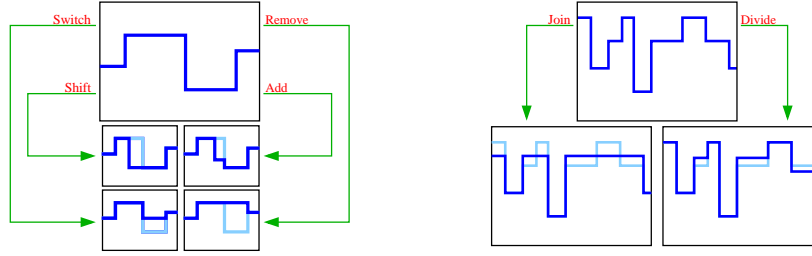

Figure 2: Example showing how the proposal actions modify the path of the Chinese restaurant process. The new path is drawn in dark blue, the old one in light blue.

While the data likelihood ratio is the same for all proposal actions and follows from (1), the proposal and prior ratios

$$\Psi = \frac{Q(\lambda_{(0:T)}|\lambda_{(0:T)}^*)}{Q(\lambda_{(0:T)}^*|\lambda_{(0:T)})} \frac{P(\lambda_{(0:T)}^*|f, \alpha, p_\lambda)}{P(\lambda_{(0:T)}|f, \alpha, p_\lambda)} \tag{7}$$

depend on the chosen proposal action. The acceptance probability for each action (provided in the supplementary material) can be calculated based on its description and the probability of a path (2).

Because our proposal process is a simple random walk, the major contribution to the computation time comes from calculating the data likelihood. Luckily, this can be done very efficiently, because we only need to know how many Poisson events occur during the segments of $\lambda_{(0:T)}^*$ and $\lambda_{(0:T)}$, how often the process changes state, and how much time it spends in each state. In order to avoid iterating over all the data for each proposal, we compute the index of the next event in the data for a fine time grid before the sampler starts. This ensures that the computational time is linear in the number of jumps in $\lambda_{(0:T)}$, while the number of Poisson events in the data only introduces one-time costs for calculating the grid, which are negligible in practice. Additionally, we only need to compute the likelihood ratio over those segments which are changed in the proposal, because the unchanged parts cancel each other out.

## 3.2 Sampling the parameters

As we use a gamma prior $\text{Gamma}(\lambda_i; a, b)$ for each $\lambda_i$, it is easy to see from (1) that this leads to gamma posteriors

$$\text{Gamma}\left(\lambda_i; a + n_i, b/(\tau_i b + 1)\right) \tag{8}$$

over $\lambda_i$. Thus a Gibbs sampling step is used to update each $\lambda_i$. As for the rate $f$ of change points, if we assume a gamma prior for $f \sim \text{Gamma}(a_f, b_f)$, the posterior becomes a gamma distribution, too:

$$\text{Gamma}\left(f; a_f + c, b_f/(T b_f + 1)\right). \tag{9}$$

# 4 Experiments

We first validate our sampler on synthetic data sets, then we test our Chinese restaurant approach on neural spiking data from a cat's primary visual cortex.

## 4.1 Synthetic Data

We sampled 100 data sets from the prior with $f = 0.02$ and $\alpha = 3.0$. Figure 3 compares the true values for the number of states and number of jumps with the posterior mean after 1.1 million samples with the first $100,000$ dropped as burn-in. On average the sampler took around 25 seconds to generate the samples on an Intel Xeon CPU with $2.40$ GHz.

The amounts of both jumps and states seem to be captured well, but for a large number of distinct states the mean seems to underestimate the true value. This is not surprising, because the $\lambda$ parameters are drawn from the same base distribution. For a large number of states the probability that two

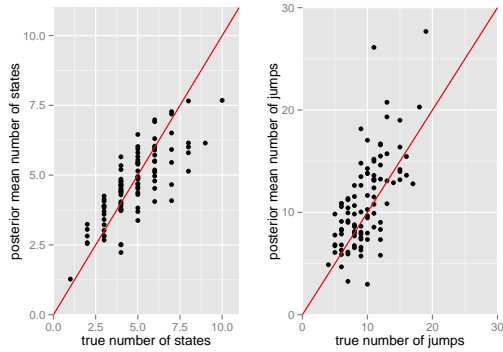

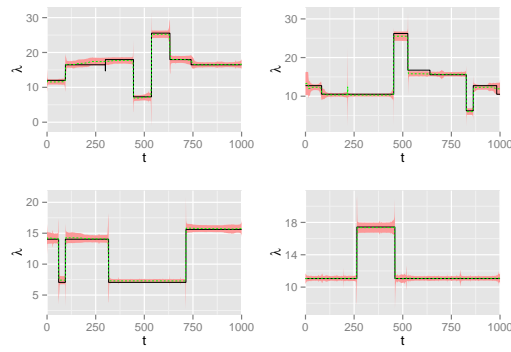

Figure 3: Posterior mean vs. true number of states (*left*) and jumps (*right*) for 100 data sets drawn from the prior. The red line shows the identity function.

Figure 4: Posterior of $\lambda$ over $t$ for the first 4 toy data sets. The black line is the true path, while the posterior mean is drawn as a dashed green lined surrounded by a 95% confidence interval.

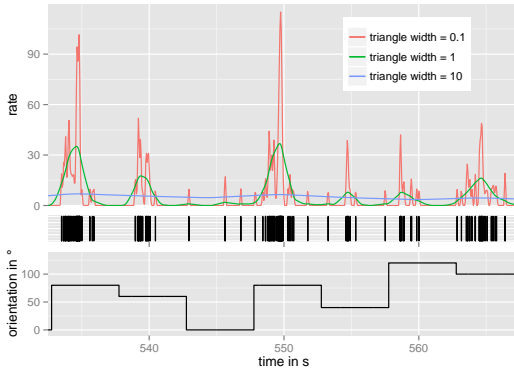

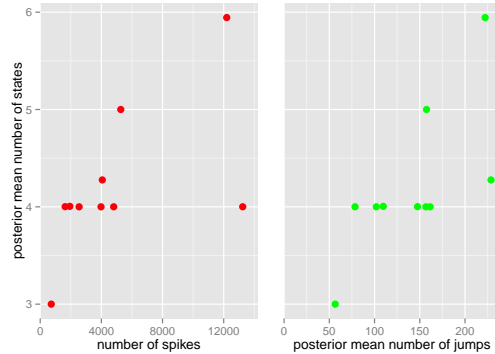

Figure 5: Stimulus and data for a part of the recordings from the first neuron. (*top*) Mean rates computed by using a moving triangle function. (*middle*) Spiking times. (*bottom*) Orientation of the stimulus.

Figure 6: (*left*) Posterior mean number of states vs. number of spikes in the data for all neurons. (*right*) Posterior mean number of states over the posterior mean number of jumps.

states are very similar becomes high, which makes them indistinguishable without observing more data. For four of the 100 data sets the posterior distribution over $\lambda(t)$ is compared to the true path in figure 4. While we used the true value of $\alpha$ for our simulations the model seems to be robust against different choices of the parameter. This is shown in the supplementary material.

## 4.2 Bursting of Cat V1 Neurons

Poisson processes are not an ideal model for single neuron spiking times [3]. The two main reasons for this are the refractory period of neurons and bursting [14]. Despite this, Poisson processes have been used extensively to analyze spiking data [e.g. 19, 20]. Additionally, both reasons should not be a problem for us. The refractory period is not as important for inference since spiking during it will not be observed. Bursting, on the other hand, is exactly what models with jumping Poisson rates are made to explain: sudden changes in the spiking rate.

The data set used in this paper was obtained from multi-site silicon electrodes in the primary visual cortex of an anesthetized cat. For further information on the experimental setup see [4]. The data set contains spike trains from 10 different neurons, which were recorded while bars of varying orientation moved through the visual field of the cat. Since the stimulus is discrete (the orientation

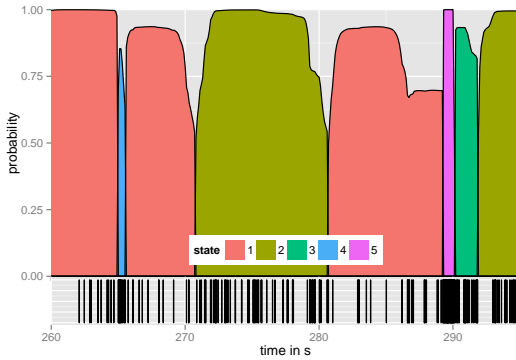

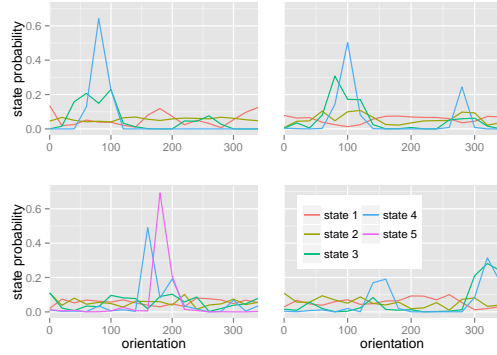

Figure 7: Detail of the results for one of the neurons. The black lines at the bottom represent the spike data, while the colors indicate the state with the highest posterior probability, which is represented by the height of the area. The states are ordered by increasing rate $\lambda$.

Figure 8: Probability distribution of the orientation of the stimulus conditioned on the active state. The states are ordered by increasing rate $\lambda$ and the results are taken from samples at the MAP number of states.

ranges from $0°$ to $340°$ in steps of $20°$), we expect to find discrete states in the response of the neurons. The recording lasted for 720 seconds and, while the orientation of the stimulus changed randomly, each orientation was shown 8 times for 5 seconds each over the whole experiment. In figure 5, a section of the spiking times of one neuron is shown together with the orientation of the stimulus. When computing a mean spiking rate by sliding a triangle function over the data, it is crucial to select a good width for the triangle function. A small width makes it possible to find short phases of very high spiking rate (so called *bursts*), but also leads to jumps in the rate even for single spikes. A larger width, on the other hand, smoothes the bursts out. Using our sampler for Bayesian inference based on our model allows us to find bursts and cluster them by their spiking rate, but at the same time the spikes between bursts are explained by one of the ground states, which have lower rates, but longer durations.

We used an exponential prior for $f$ with mean rate $10^{-4}$ and a low value of $\alpha = 0.1$ to prevent overfitting. A second simulation running with a ten times higher prior mean for $f$ and $\alpha = 0.5$ lead to almost the same posterior number of states and only a slightly higher number of jumps, of which a larger fraction had no impact, because the state was not changed. The base distribution $p_\lambda$ was chosen to be exponential with mean $10^6$, which is a fairly uninformative prior, because the duration of a single spike is in the order of magnitude of $1\mathrm{ms}$ [11] resulting in an upper bound for the rate at around $1000/\mathrm{s}$.

The posterior number of states for all of the 10 neurons is in the same region, as shown in figure 6, even though the number of spikes differs widely (from 725 to 13244). Although there seem to be more states if more jumps are found, the posterior differs strongly from the prior—a priori the expected number of states is under 2—indicating that the posterior is dominated by the data likelihood.

For a small time frame of the spiking data from one of the neurons figure 7 shows which state had the highest posterior probability at each time and how high this probability was. It can be seen that the bursting states, which have high rates, are only active for a short time. Figure 8 shows that these burst states are clearly orientation dependent (see the supplementary material for results of all 10 neurons). Over the whole experiment all orientations were shown for exactly the same amount of time. While the highest state is always clearly concentrated on a range of about $60°$, the lower bursting states cover neighboring orientations. Often a smaller reaction can be seen for bars rotated by $180°$ from the favored angle. The lowest state might indicate inhibition, because it is mostly active between the favored state and the one rotated by $180°$.

As we can see in figure 9, some of the rates of the states are pretty similar over all the neurons, although it has to be noted that the orientation is probably not the only feature of the stimulus the

neurons are receptive to. Especially the position of the bar in the visual field should be important and could explain, why only some of the neurons reach the highest burst rate.

It may seem that finding bursts is a simple task, but there has been extensive work in this field [e.g. 6, 13, 16] and naive approaches, like looking at the mean rate of events over time, fail easily, if the time resolution is not chosen well (as seen in figure 5). Additionally, our sampler not only distinguishes between burst and non-burst phases, but also uncovers discrete intensities, which are associated with features of the stimulus.

### 4.3 Comparison to a continuous rate model

While our model assumes that the Poisson rates are discrete values, there have been other approaches applying continuous functions to estimate the rate. [1] use a Gaussian process prior over $\lambda(t)$ and present a Markov chain Monte Carlo sampler to sample from the posterior. Since the sampler is very slow for our neuron data, we restricted the inference task to a small time window of the spike train from only one of the neurons.

In figure 10 the results from the Sigmoidal Gaussian Cox Process (SGCP) model of [1] are shown for different values of the length scale hyperparameter and contrasted with the results from our model. Similar to the naive approach of computing a moving average of the rate (as in figure 5) the GP seems to either smooth out the bursts or becomes so sensitive that even single spikes change the rate function significantly depending on the choice of the GP hyperparameters.

Our neural data seems to be especially bad for the performance of this algorithm, because it is based on the principle of uniformization. Uniformization was introduced by [9] and allows to sample from an inhomogeneous Poisson process by first sampling from a homogeneous one. If the rate of the homogeneous process is an upper bound of the rate function of the inhomogeneous Poisson process, then a sample of the latter can be generated by thinning out the events, where each event is omitted with a certain probability. The sampler for the SGCP model performs inference using this method, so that events are sampled at the current estimate of the maximum rate for the whole data set and thinned out afterwards.

For our neural data the maximum rate would have to be the spiking rate during the strongest bursts, but this would lead to a very large number of (later thinned out) event times to be sampled in the long periods between bursts, which slows down the algorithm severely. This problem only occurs if uniformization is applied on $\lambda(t)$ while other approaches, like [21], use it on the rate of a MJP with a fixed number of states.

When we use a practically flat prior for the sampling of the maximum rate, it will be very low compared to the bursting rates our algorithm finds (see figure 10). On the other hand, if we use a very peaked prior around our burst rates, the algorithm becomes extremely slow (taking hours for just 100 samples) even when used on less than a tenth of the data for one neuron.

## 5  Conclusion

We have introduced an inhomogeneous Poisson process model with a flexible number of states. Our inference is based on a MCMC sampler which detects recurring states in the data set and joins them in the posterior. Thus the number of distinct event rates is estimated directly during MCMC sampling.

Clearly, sampling the number of states together with the jump times and rates needs considerably more samples to fully converge compared to a MJP with a fixed number of states. For our application to neural data in section 4.2 we generated 110 million samples for each neuron, which took between 80 and 325 minutes on an Intel Xeon CPU with 2.4 GHz. For all neurons the posterior had converged at the latest after a tenth of the time. It has to be remembered that to obtain similar results without the Chinese restaurant process, we would need to compute the Bayes factors for different number of states. This is a more complicated task than just doing posterior inference for a fixed number of states and would require more computationally demanding approaches, e.g. a bridge sampler, in order to get reasonably good estimates. Additionally, it would be hard to decide for what range of state dimensionality the samplers should be run. In contrast to this, our sampler typically gave a good estimate of the number of states in the data set already after just a few seconds of sampling.

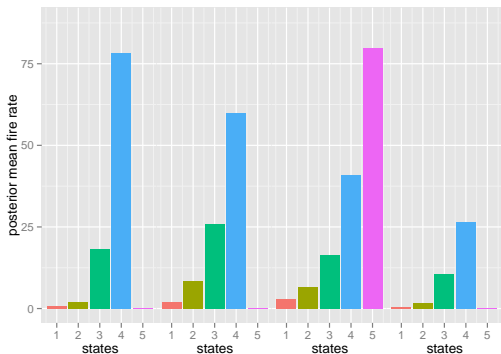
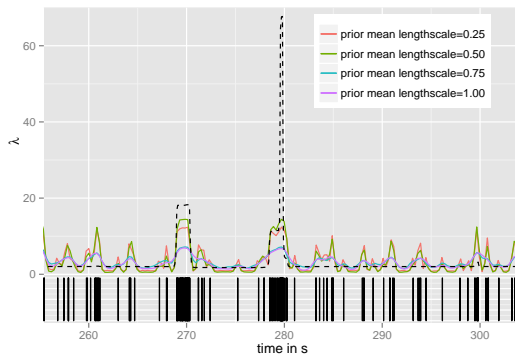

Figure 9: Posterior mean rates $\lambda_i$ for the MAP number of states.

Figure 10: Results of the SGCP Sampler on a small part of the data of one neuron. The black dashed line shows the posterior mean from our sampler. The spiking times are drawn as black vertical lines below.

Longer run times are only needed for a higher accuracy estimate of the posterior distribution over the number of states.

Although our prior for the transition rates of the MJP is state-independent, which facilitates the integration over the maximum number of states and gives rise to the Chinese restaurant process, this does not hold for the posterior. We can indeed compute the full posterior state transition matrix—with state-dependent jump rates—from the samples.

A huge advantage of our algorithm is that its computation time scales linearly in the number of jumps in the hidden process and the influence of the number of events can be neglected in practice. This has been shown to speed up inference for MMPPs [23], but our more flexible model makes it possible to find simple underlying structures in huge data sets (e.g. network access data with millions of events) in reasonable time without the need to fix the number of states beforehand.

In contrast to other MCMC algorithms [2, 8, 15] for MMPPs, our sampler is very flexible and can be easily adapted to e.g. Gamma processes generating the data or semi-Markov jump processes, which have non-exponentially distributed waiting times for the change of the rate. For Gamma process data the computation time to calculate the likelihood would no longer be independent of the number of events, but it might lead to better results for data which is strongly non-Poissonian.

We showed that our model can be applied to neural spike trains and that our MCMC sampler finds discrete states in the data, which are linked to the discreteness of the stimulus. In general, our model should yield the best results when applied to data with many events and a discrete structure of unknown dimensionality influencing the rate.

## Acknowledgments

Neural data were recorded by Tim Blanche in the laboratory of Nicholas Swindale, University of British Columbia, and downloaded from the NSF-funded CRCNS Data Sharing website.

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
