[Supplementary Material]

# Supplementary Material: Poisson Process Jumping between an Unknown Number of Rates: Application to Neural Spike Data

**Florian Stimberg**
Computer Science, TU Berlin
Florian.Stimberg@tu-berlin.de

**Andreas Ruttor**
Computer Science, TU Berlin
Andreas.Ruttor@tu-berlin.de

**Manfred Opper**
Computer Science, TU Berlin
Manfred.Opper@tu-berlin.de

## 1   Sampler: Acceptance probabilities

We accept a path $\lambda^*_{(0:T)}$ with probability

$$p_{\text{MH}} = \min\left(1, \frac{P(\mathbf{Y}|\lambda^*_{(0:T)})}{P(\mathbf{Y}|\lambda_{(0:T)})}\frac{Q(\lambda_{(0:T)}|\lambda^*_{(0:T)})}{Q(\lambda^*_{(0:T)}|\lambda_{(0:T)})}\frac{P(\lambda^*_{(0:T)}|f,\alpha,p_\lambda)}{P(\lambda_{(0:T)}|f,\alpha,p_\lambda)}\right). \tag{1}$$

The data likelihood ratio remains the same for all proposal actions and follows from

$$P(\mathbf{Y}|\lambda_{(0:T)}) \propto \prod_{i=1}^{s}\lambda_i^{n_i}e^{-\tau_i\lambda_i}. \tag{2}$$

On the other hand, the proposal and prior ratios

$$\Psi = \frac{Q(\lambda_{(0:T)}|\lambda^*_{(0:T)})}{Q(\lambda^*_{(0:T)}|\lambda_{(0:T)})}\frac{P(\lambda^*_{(0:T)}|f,\alpha,p_\lambda)}{P(\lambda_{(0:T)}|f,\alpha,p_\lambda)} \tag{3}$$

depend on what proposal action was chosen and can be obtained from the prior probability of a path

$$P(\lambda_{(0:T)}|f,\alpha,p_\lambda) \propto f^c e^{-fT}\alpha^s \frac{\prod_{j=1}^{s}(p_\lambda(\lambda_j)(\#_j-1)!)}{\prod_{i=0}^{c}(\alpha+i)} \tag{4}$$

and the description of the action.

For **shifting** the time of a jump it is

$$\Psi = \frac{\Phi((t_{\max}-t)/\sigma_t)-\Phi((t_{\min}-t)/\sigma_t)}{\Phi((t_{\max}-t^*)/\sigma_t)-\Phi((t_{\min}-t^*)/\sigma_t)}, \tag{5}$$

where $t^*$ is the proposed new time, $\sigma_t$ the standard deviation of the Gaussian and $\Phi(\cdot)$ is the cumulative distribution function of the standard normal distribution.

When **adding** a jump, we distinguish between adding a new value $\lambda_{s+1}$ and reusing an existing one. In the first case the ratio is

$$\Psi = \frac{q_r T}{q_a q_n(c+1)\gamma^*(\lambda_{s+1})}\frac{f\alpha}{(\alpha+c+1)}, \tag{6}$$

with $q_n$ as the proposal probability to add a new value $\lambda_{s+1}$ and $\gamma^*(\lambda_i)$ being the gamma density at $\lambda_i$ with shape $n_i^*+1$ and inverse scale $\tau_i^*$. This is proportional to the likelihood of the data given the parameter $\lambda_i$ and the new path $\lambda^*_{(0:T)}$.

When we do not add a new state and instead reuse $\lambda_i$, we get

$$\Psi = \frac{q_r T}{q_a(1-q_n)(c+1)p_{\text{seg}}^*(i)} \frac{f(\#_i^* - 1)}{(\alpha + c + 1)}, \tag{7}$$

where $\#_i$ is the number of segments which use $\lambda_i$ in the old path and $p_{\text{seg}}^*(i)$ denotes the probability to choose $\lambda_i$ for the segment (see section 2 for details).

When we **remove** a jump, we distinguish between removing a jump and thereby removing the last instance of a state $i$ ($\#_i^* = 0$):

$$\Psi = \frac{q_a q_n c \gamma(\lambda_i)}{q_r T} \frac{(\alpha + c)}{f\alpha}, \tag{8}$$

and the case that the state is still used after removing the jump ($\#_i^* > 0$):

$$\Psi = \frac{q_a(1-q_n)cp_{\text{seg}}(i)}{q_r T} \frac{(\alpha + c)}{f(\#_i - 1)}. \tag{9}$$

When we **switch** the value of $\lambda$ used in a segment from $\lambda_i$ to $\lambda_j$ we have four cases:

1. $\lambda_i$ is still used in the proposal ($\#_i^* > 0$) and $\lambda_j$ is already assigned to another segment ($\#_j > 0$):

$$\Psi = \frac{p_{\text{seg}}(i)}{p_{\text{seg}}^*(j)} \frac{(\#_j^* - 1)}{(\#_i - 1)}. \tag{10}$$

2. $\lambda_i$ is still used in the proposal ($\#_i^* > 0$) and we introduce a new value $\lambda_j$ ($\#_j = 0$):

$$\Psi = \frac{(1-q_n)p_{\text{seg}}(i)}{q_n \gamma^*(\lambda_j)} \frac{\alpha}{(\#_i - 1)}. \tag{11}$$

3. $\lambda_i$ is no longer used in the proposal ($\#_i^* = 0$) and $\lambda_j$ is already used in another segment ($\#_j > 0$):

$$\Psi = \frac{q_n \gamma^*(\lambda_i)}{(1-q_n)p_{\text{seg}}^*(j)} \frac{(\#_j^* - 1)}{\alpha}. \tag{12}$$

4. $\lambda_i$ is no longer used in the proposal ($\#_i^* = 0$) and we introduce a new value $\lambda_j$ ($\#_j = 0$):

$$\Psi = \gamma(\lambda_i)/\gamma^*(\lambda_j). \tag{13}$$

**Joining** two neighboring states $i_1$ and $i_2$ into a new state $j$ leads to

$$\Psi = \frac{q_d p_{\text{par}} p_\epsilon(\epsilon)(s-1)}{q_j s_{>1}^*} \frac{\epsilon}{2\lambda_j} \frac{p_\lambda(\lambda_j^*)(\#_j^* - 1)!}{\alpha p_\lambda(\lambda_{i_1})p_\lambda(\lambda_{i_2})(\#_{i_1} - 1)!(\#_{i_2} - 1)!}, \tag{14}$$

where $q_d$ and $q_j$ are the probabilities to choose the *divide* and *join* action, respectively, $p_\epsilon$ is the density from which the factor $\epsilon > 1$ is drawn and $p_{\text{par}}$ is the probability to assign the segments between states $i_1$ and $i_2$ like they are in the original path (see section 2 for details). $s_{>1}^*$ is the number of states in the proposal with more than one segment assigned. $\epsilon/(2\lambda_j)$ is a Jacobian factor resulting from using the distribution over $\epsilon$ rather than that of the $\lambda$-values when applying the join and divide actions.

For **dividing** state $i$ into the new states $j_1$ and $j_2$ we get

$$\Psi = \frac{q_j s_{>1}}{q_d p_{\text{par}} p_\epsilon(\epsilon)s} \frac{2\lambda_i}{\epsilon} \frac{\alpha p_\lambda(\lambda_{j_1}^*)p_\lambda(\lambda_{j_2}^*)(\#_{j_1}^* - 1)!(\#_{j_2}^* - 1)!}{p_\lambda(\lambda_i)(\#_i - 1)!}. \tag{15}$$

## 2 Sampler: Assigning a $\lambda$ value to a segment

If we reuse an existing state when **adding** a jump or **switching** the state of a segment we choose the new state $i$ randomly with probability proportional to

$$P(\lambda_i|\mathbf{Y}, \lambda_{(0:T)}) = \text{Gamma}\left(\lambda_i; a + n_{seg}, \frac{b}{\tau_{seg}b + 1}\right), \tag{16}$$

Figure 1: Robustness of the posterior to different values of $\alpha$ for four of the toy datasets.

where $n_{seg}$ is the number of Poisson events during the segment, $\tau_{seg}$ is the width (in time units) of the segment and $a$ and $b$ come from the base distribution $p_\lambda(\lambda) = \mathrm{Gamma}(\lambda; a, b)$.

Therefore the probability to choose state $i$ becomes

$$p_{seg}(i) = \frac{\lambda_i^{a+n_{seg}-1} \exp\left(-\frac{\tau_{seg}b+1}{b}\lambda_i\right)}{\sum_{j=1}^{s} \lambda_j^{a+n_{seg}-1} \exp\left(-\frac{\tau_{seg}b+1}{b}\lambda_j\right)}. \tag{17}$$

After **dividing** a state $i$ into two new states $j_1$ and $j_2$ the segments of the original state must be assigned to the new states. For segment $l$ the probability to be assigned to state $j_1$ is

$$p'_{seg}(l, j_1) = \frac{\lambda_{j_1}^{a+n_l-1} \exp\left(-\frac{\tau_l b+1}{b}\lambda_{j_1}\right)}{\lambda_{j_1}^{a+n_l-1} \exp\left(-\frac{\tau_l b+1}{b}\lambda_{j_1}\right) + \lambda_{j_2}^{a+n_l-1} \exp\left(-\frac{\tau_l b+1}{b}\lambda_{j_2}\right)}, \tag{18}$$

and accordingly for state $j_2$. Let $\omega_1 \ldots \omega_{\#_i} \in \{j_1, j_2\}$ be the assignments of the $\#_i$ segments of state $i$ to the new states then we get

$$p_{par} = \prod_{l=1}^{\#_i} p'_{seg}(l, \omega_l). \tag{19}$$

If one of the new states is assigned to all segments but the last one, then the last segment is automatically assigned to the other state thereby setting $p_{seg}(\#_i, \omega_{\#_i}) = 1$.

## 3 Choice of $\alpha$ Parameter

Our sampler used the same $\alpha$ value that was used to generate the data to get the results on the 100 datasets. For the first 4 datasets we let the sampler run with a wide range of different values of $\alpha$ and, as can be seen in figure 1, the model is robust enough to cope with this. This means it is sufficient if we have a broad idea of the value of $\alpha$ and therefore the numbers of states to expect.

## 4 Further Results for Neuron Data

Figure 2 shows the connection between the states and the orientation of the stimulus for all neurons in the dataset. The data contained spike trains from 10 neurons, but we kept the original non-consecutive numbering.

Figure 2: Probability distribution over the orientation of the stimulus while a state is active. States are ordered by increasing $\lambda$ rate. All results at the maximum a posteriori number of states.