[Reviews · NeurIPS 2014]

Submitted by Assigned_Reviewer_11

In reply to the author's feedback "Our description of uniformization might be misleading, because the problems we describe do not occur in all of its applications. For the SGCP model discussed in our paper, however, the uniformization really is over the rate, which is lambda in our model. There is no MJP in the SGCP model, because the rate is continuous." After rereading the relevant sections of the paper, I am sure that this is incorrect. From page 2 line 070 "in our model lambda(t) is a latent piecewise constant process. ... Let us assume we already have i segments [from the piecewise constant functino lambda(t)] and draw the next jump time from an exponential distribution with rate f".

Therefore, the authors' process is an MJP with constant rate f, and state space given by the positive reals. The emission model of the MJP is a Poisson process with rate lambda(t). To use uniformization in this case, we must upper bound the rate of the transitions of the MJP --- that is done by chosing f. Surely, lambda(t) might be large during bursts, but f is still constant and so the number of auxiliary uniformization events does not depend on lambda(t).

"For our neural data the maximum rate would have to be the spiking rate during the strongest
bursts, but this would lead to a very large number of (later thinned out) event times to be sampled in the long periods between bursts, which slows down the algorithm severely." That is incorrect. The maximum rate used by uniformization is f (that's the transition rate of the latent piecewise linear process) which is constant and doesn't depend on the spike rate.

---

In the manuscript "Poisson Process Jumping between an Unknown Number of Rates: Application to Neural Spike Data" the authors provide a model for spike train data using an Markov jump process (MJP) modulating the rate of spikes. The state space of the MJP is given by a Dirichlet process (DP) supported on the rate of spikes. The DP is then marginalized, and Metropolis-Hastings updates are derived for the resulting CRP. The authors apply this model to simulated data and also bursting in V1 neurons of cats (data from NSF-funded CRCNS Data Sharing repository). Overall, I thought that the manuscript was good. I did however think that it could have made more contact with computational neuroscience literature. Also, the manuscript incorrectly describes uniformization and the implications for efficiency.

The authors' approach is good: it applies modern machine learning to a very concrete problem. The manuscript is therefore topical for the Neural Information Processing Systems conference. However, I felt like this paper could have been more in touch with classical work on HMMs in computational neuroscience. For example, I felt that the authors should have extended their model to ensemble cases. HMM methods have long been used in computational neuroscience to model spike rates (see Abeles, M., & Gerstein, G. L. 1988. Detecting spatiotemporal firing patterns among simultaneously recorded single neurons. Journal of Neurophysiology, 60(909), 909–24. for example.) Rather than focusing on single neurons, each HMM state in Abeles & Gerstein 1988 is parameterized by a separate rate for each neuron in the ensemble. Almost every other work on HMMs in computational neuroscience also looks at ensemble cases.

The same model presented . was used in Vinayak Rao's PhD thesis for modelling neural spikes (chapter 5.4.4 of MCMC for continuous-time discrete-state systems, University College London, 2012). In that work, a Dirichlet process prior was put on the spike rates of neurons in a grasshopper auditory cortex, and then the rates and transitions were learned using uniformization. Note that this work was not published outside of his thesis and also his thesis does not go into much detail about the paradigms used, so I would definitely not be opposed to anyone publishing a detailed account of it.

The authors do mention uniformization as another approach to this sort of data (page 7 lines 351-365), but they state that it would be inefficient because uniformization requires an upper bound on the rate, and this upper bound would be large due to bursting. Unfortunately, this is a misunderstanding of the way uniformization works and it's not the case that uniformization would be inefficient, at least not for this reason. The upper bound required in uniformization is on the transition rate of the underlying MJP, not on the likelihood parameter (the spike rate). The author's model, viewed as an MJP, has transition rate f, which is a parameter they mention on page 2 line 78. Because f is constant over the process, an upper bound on the transition rate of the MJP is given by f, and this (not the spike rate) would be used as an upper bound in uniformization. I do agree that it is confusing because there are two rates here: (1) the transition rate of the latent piecewise linear function (which is f) and (2) the spike rate (which is lambda(t)), but the upper bound in uniformization cares only about (1) and does not care about (2).

I found that the comment by the authors that the shifting, adding, removing and switching updates alone were "very unlikely to reduce the number of states" to be sort of strange. If those actions are balanced, then around a mode shouldn't they be just as likely to reduce the number of states as to increase the number of states? Is the issue that these updates alone won't mix and therefore won't get near a mode? Would be nice to make that more explicit.

Another comment that I found sort of strange was "The amounts of both jumps and states seem to be captured well, but for a large number of distinct states the mean seems to underestimate the true value. This is not surprising, because the lambda parameters are drawn from the same base distribution. For a large number of states the probability that two states are very similar becomes high, which makes them indistinguishable without observing more data" (page 5 lines 256 to 260) which the authors wrote to explain why the emperical statistic of the posterior number of states in Figure 3 (left) falls below the y=x line in the qq-plot. While it is true that the model will be unable to distinguish between lambda values that are close together with a small amount of data, it is also true that due to sampling noise (MCMC variance) a sampler will also occasionally put data that is close together and well explained by one cluster into two (or more) separate cluster, and the rate at which this happens should sort of cancel out the rate of `incorrectly' clustering data from more than one (very similar) clusters into one cluster. Due to the main theorem behind MCMC, under the procedure: 1. sample parameters y from the prior, 2. sample data x|y, 3. sample Y|x through your MCMC scheme, the variable Y should be an unbiased estimator of y. So, unless I am missing something, it seems the mismatch in the qq-plot in Figure 3 (left) is still unexplained and could indicate failure to mix, or a bug.

Minor notes:
The citations within the text of this manuscript are not conforming to the NIPS standard. NIPS should cite references using [1], [2], [3] etc, and then have [1], [2], [3] preceeding the citations in the reference section. While this might seem pedantic, it should be changed for the camera ready copy should the paper be accepted, in order to allow the entire NIPS proceedings to maintain it's well deserved familiar and professional feeling.
Summary: Application to Neural Spike Data" the authors provide a model for spike train data using an Markov jump process (MJP) modulating the rate of spikes. The state space of the MJP is given by a Dirichlet process (DP) supported on the rate of spikes.

Submitted by Assigned_Reviewer_14

The authors introduce a non-parametric inhomogeneous Poisson process to model neural spike data, where the rate is assumed to be piecewise constant and represented by a Chinese restaurant process. This allows for the Poisson process to switch between states without requiring a fixed number of states ahead of time. A Markov chain Monte Carlo sampler is described, providing shift, add, remove and switch moves to allow sufficient mixing of the chain. The methods are implemented on both synthetic and real data.

The idea of piecewise constant Poisson processes is not new, the novelty lies in that the states are modelled through a CRP. The paper is generally well written but more thorough comparisons would be beneficial. The superiority of the proposed methods against state-of-the-art is not sufficiently addressed and motivated. For example, there is no formal comparative model-assessment on the real dataset other than heuristics based on the plots.

A few minor comments:
* When merging two values of lambda, a deterministic function is used for lambda*. Why is that beneficial compared to a random proposal that uses p(lambda* | Y, lambda)?
* There could be some discussion on how this could be extended to higher dimensions.
* In formula (7) (and some related formulae), the two subscripts 0:T do not look identical (because of the *) and it makes the expression a little harder to read.
* The Poisson process assumes independent increments. Is this a realistic assumption in this context?
* Typos: line 362 "an"-->"a", line 398 "then"-->"than"
* The number of states in the presented example is rather low. In this case a standard DP would also have worked (eg truncated at a large number of states). How would it compare against the proposed analysis?

Summary: The idea in the paper is useful and could be applicable to a wide variety of data, although other piecewise approaches exist. However, the model at hand hasn't been thoroughly investigated.

Submitted by Assigned_Reviewer_25

The authors introduce a model where the rate of an inhomogeneous Poisson process is characterized through a Chinese Restaturant Process. The idea is quite original and the results quite promising. Unfortunately, the paper is also quite poorly written, which takes away some interest. Some notes below:

l. 72: what's T? What's a path $\lambda_{(0:T)}$? They have never been defined/described before. Analogously, l. 73: what's Y? What's s?

l. 76 "$\tau_i$ is the overall time spent in state $s_i$ defined as $\lambda(t)=\lambda_i$"...grammatically odd

l. 78 What's c? How is it chosen? Is there a prior on it?

l. 80 Not sure how the Dirichlet Process comes about. What's $\pi$?

Note: I am not saying that I don't know what it is, I am saying that someone who doesn't already understand what you want to do will never guess what you are trying to say.

line 110: please, specify what parameters are held fixed

In figure 4, the true values of $\lambda$ appear quite well separated. What happens if that's not the case?

line 377 Why do you need a BF? What model are you thinking at for this comparison?

line 401 If you already get a good estimate of the number of states after a few seconds, the next defauls question is "how's the mixing of the chain?"

Summary: The paper has an original and interesting contribution. However, greater care should be taken in the presentation of the model and interpretation of the results.
Author Feedback
Author rebuttal: We thank all reviewers for their comments.

@Reviewer11:

We will cite Rao's thesis and change the citation style in the final paper.

Our description of uniformization might be misleading, because the problems we describe do not occur in all of its applications. For the SGCP model discussed in our paper, however, the uniformization really is over the rate, which is lambda in our model. There is no MJP in the SGCP model, because the rate is continuous.

Add and switch are able to add a new state with only one move, while several remove or switch actions are necessary to reduce the number of states if the state is used in multiple segments.

About Figure 3: The (left) plot shows the posterior expectation of the number of states vs exact number of states. Conditioned on the exact number of states the former expectation will NOT be an unbiased estimate of the latter. Take e.g. the limit of vanishing number of observations. In this case, the posterior mean equals the prior mean which is a constant. The problem with a large number of states compared to the prior is as follows: the likelihood is almost equal if the rates are similar, but the prior probability of a higher number is clearly smaller. Therefore the posterior favors a smaller number of states. There can be extreme cases like two states with rates so low that both do not produce any events.

@Reviewer14:

The deterministic choice of lambda after joining states is needed for reversibility, because otherwise we could draw a value which would not allow us to draw the old values of lambda in our divide action.

In the beginning of section 4.2 we talk about the fit of the Poisson assumption to neural data.

A standard DP with a truncated number of states might work, but the maximum number would have to be chosen beforehand. However, the sampler would be inefficient if it was chosen too high, and not able to give the correct estimate if it was chosen too low.

@Reviewer25:

We will make the definitions clearer in the final version.

c is drawn from a Poisson distribution with rate f, which has a Gamma prior.

If the lambda rates are not well separated and the corresponding segments are short, they might be interpreted as one rate. This happens in some samples in the first dataset in figure 4 between t~100 and t~400.

We would need Bayes factors if we had a fixed number of states and we would compare models where that number is different. This was for example done in Chris Sherlock's PhD thesis.

The chain is quick to go to the region of high probability, but changes in the number of states are not accepted as often as other moves.